# An anchor in troubled times: Trust in science before and within the COVID-19 pandemic

**Rainer Bromme** [1]*, **Niels G. Mede** [2], **Eva Thomm**[3], **Bastian Kremer**[4], **Ricarda Ziegler** [4]

**1** Department of Psychology, University of Münster, Münster, Germany, **2** Department of Communication and Media Research (IKMZ), University of Zürich, Zürich, Switzerland, **3** Faculty of Education, University of Erfurt, Erfurt, Germany, **4** Wissenschaft im Dialog (WiD), Berlin, Germany

* bromme@uni-muenster.de

**Data Availability Statement:** The analyses could be reproduced with the R syntax, which we share publicly with the complete survey data and original questionnaires in the Open Science Framework repository (osf.io/czn4g).

## Abstract

Researchers, policy makers and science communicators have become increasingly been interested in factors that affect public's trust in science. Recently, one such potentially important driving factor has emerged, the COVID-19 pandemic. Have trust in science and other science-related beliefs changed in Germany from before to during the pandemic? To investigate this, we re-analyzed data from a set of representative surveys conducted in April, May, and November 2020, which were obtained as part of the German survey *Science Barometer*, and compared it to data from the last annual *Science Barometer* survey that took place before the pandemic, (in September 2019). Results indicate that German's trust in science increased substantially after the pandemic began and slightly declined in the months thereafter, still being higher in November 2020 than in September 2019. Moreover, trust was closely related to expectations about how politics should handle the pandemic. We also find that increases of trust were most pronounced among the higher-educated. But as the pandemic unfolded, decreases of trust were more likely among supporters of the populist right-wing party AfD. We discuss the sustainability of these dynamics as well as implications for science communication.

## Introduction

The COVID-19 pandemic is a new challenge for people all over the world, and it is one that can only be understood and managed based on scientific knowledge (meant in the general sense of *Wissenschaft*, which includes social sciences and humanities, [1]). As such, successfully coping with the pandemic is probably related to the general public's trust in science [2–11]. In situations like the COVID-19 pandemic, prevention measures, (such as rules on social distancing), are justified by referring to scientific evidence. Therefore, to improve the communication of such science-based measures, it is important to understand the conditions related to citizens' trust in science [12].

Of course, COVID-19 is not the first global challenge that must be understood and managed based on scientific knowledge; the utmost example being climate change. Therefore, it is highly important to learn about the extent and the conditions of public trust in science when

**Funding:** This paper makes use of the he annual German Science Barometer as well as its Corona Special edition survey which was done by Wissenschaft im Dialog (WiD). The Science Barometer project is funded and supported by the Robert Bosch Stiftung (00906101-0030) and the Fraunhofer-Gesellschaft. The funders of the survey data collection had no role in study design, data collection and analysis, decision to publish, or preparation of the manuscript.

**Competing interests:** The authors have declared that no competing interests exist.

facing global challenges which would help to understand how trust in science might affect a nation's resilience against crises [13–15].

The pandemic is dynamic, and its spread as well as related containment measures varied between countries and over time. Overall, the German public is politically less polarized compared to many countries, and, the number of COVID-19 infections (and as well the COVID-19-related mortality) has been smaller than in most other countries in Europe. Moreover, especially at the beginning of the pandemic, the containment of the outbreak was rather successful. Nevertheless, these early successes melted away in late 2020 as the pandemic endured. We report on the situation in Germany as an interesting case (and hope that this will provide motivation for a series of comparable reports in other countries).

Arguably, the most established approach to measuring public opinion about science is to conduct large-scale population surveys (e.g., [16]). However, most available COVID-19-related opinion surveys have focused on public's perception of the pandemic and of the protection measures; trust in science has often only been briefly touched on [17–27], (however see [2,3,8,9] for a focus on trust in science as a predictor of protecting behavior). Typically, trust in science is measured comparatively as trust in the information provided by institutions such as universities versus health authorities. In the context of COVID-19, trust is conceptually understood as a condition or predictor of how people perceive the pandemic and whether they accept the related protection measures. We contribute to these approaches by examining how the pandemic has affected the variability of trust in science.

Surveys related to COVID-19 were, naturally, launched after the pandemic started, meaning they only provide limited insights into how public views changed from before to during the pandemic (however, see [28–30]). Yet, in several countries citizens are surveyed regularly about their attitudes to science and technology (Italy, the United States Switzerland, Sweden, and Germany, the last three surveys are called *Science Barometers*). Such established, regular surveys provide a helpful framework for scrutinizing public's trust in science before and then during the COVID-19 pandemic. Based on such a series of surveys, we view the degree of trust in science during the COVID-19 pandemic in Germany reporting the status quo of citizens' views on science at mid-April 2020, at the end of May 2020, and at the beginning of November 2020. Importantly, and unlike most existing research on public views of the pandemic, we compare these perceptions with data gathered prior to the COVID-19 outbreak, in September 2019.

We assume that the COVID-19 pandemic will have long-term impacts not only on societal institutions and citizens' everyday lives, but also on the public's belief systems related to science [31]. Therefore, our focus on trust in science will be complemented by portraying further science-related beliefs.

To assess public trust in science in Germany before and during the pandemic, we utilize survey data from the annual representative German *Science Barometer survey* (www.sciencebarometer.com). This was established by the German non-profit science communication organization *Wissenschaft im Dialog* (WiD, Science in Dialogue) in 2014. Using this data enables us to investigate public trust in science and further science-related beliefs *before* and *during* the pandemic as well as temporal changes thereof. Moreover, we can explore potential predictors of public trust and its changes. Specifically, we rely on data from four cross-sectional surveys in September 2019 (PreCOVID 09/19), April 2020 (COVID 04/20), May 2020 (COVID 05/20) and November 2020 (COVID 11/20).

We pursue two different research questions, the first being, **What are public perceptions of science, especially of trust in science, before and during the pandemic in Germany, and how have they changed (RQ 1)?** We analyze *trust in science and research* based on an item that is regularly presented in the survey. In addition, we complement this analysis by

considering further items that measure the public's general perspectives on science. Some of these items were used repeatedly throughout the surveys, while others have only been captured within some of them. Taken together, they provide a rich picture of the public's views on science and their temporal variability from before the pandemic until November 2020.

The second research question is as follows: **What are the explanatory factors of public trust in science and its variability (RQ 2)?** We examine this question using multiple regression analyses with the previously mentioned *trust in science and research* item as a criterion and with demographic variables and science-related beliefs as covariates (in a technical sense, as predictors). Reporting the RQ 2 results includes two parts. First, we look at any predictors that have been surveyed repeatedly, initially focusing on each survey period separately. Second, we examine differences in their explanatory power between subsequent survey periods. This allows us to inspect the temporal variability of the impact of these predictors of trust. The second part of reporting on RQ 2 represents a "zooming in" on predictors that had only been surveyed once, putting together an even richer picture of the possible correlates of trust in science.

Overall, to answer both research questions we rely on data from four *Science Barometer* survey waves, undertaking exploratory drillings in the terrain of science-related beliefs held by German citizens before and during the pandemic.

## Context: The COVID-19 situation in Germany in mid-April, the end of May and the beginning of November 2020

To contextualize our analysis, we first provide a summary of the COVID-19 pandemic situation in Germany during the period under scope, see [32] for a more detailed report. According to reports from the Robert Koch Institute (RKI), which is the German federal institute for public health, there was an average of 3,479 new COVID-19 cases per day in the week before the first COVID survey in April (08–14 April 2020) [33]. The average number of deaths per day was 199, and the total number of deaths had increased to 3,569. In the week before the second COVID survey in May (18–24 May 2020), there was an average of 553 new COVID-19 cases per day, an average number of deaths per day of 46, and a total number of deaths of 8,302. In the week before the November COVID survey (26 October– 01 November 2020), there was an average of 15,308 new COVID-19 cases per day, an average number of deaths per day was 68, and the total number of deaths had risen to 10,661. Accordingly, the April and November surveys took place when the so-called first and second waves of the pandemic had started to rise in Germany.

Since March 2020, the German authorities enacted numerous measures to contain the pandemic, although differences existed between nationally imposed restrictions and rules employed on a state level. Nationwide, events with more than 1,000 participants were prohibited on April 15[th]. Numerous shops, service providers in areas of personal hygiene and gastronomy were closed, as were schools and daycare centers. Additionally, the number of people who could meet in private and public contexts was restricted. However, unlike in Italy, individual outdoor activities were always allowed in Germany, and there were no general factory shutdowns.

At the end of April 2020, smaller businesses were allowed to re-open. On May 6[th], the government began easing some measures although contact restrictions were extended until June 5[th]. In general, the federal states were largely given the responsibility for repealing regulations resulting in an increasingly heterogeneous pattern of mandatory rules, recommendations, and withdrawals of restrictions. Due to the rapidly increasing numbers of new COVID-19 cases that started at the beginning of October 2020, the German government tightened measures

again with the so-called "wave-breaker lockdown". These November 2020 restrictions, portrayed as "lockdown light", were only established after several weeks of debates between the federal government and the federal states. Overall, the federal stance had been to implement earlier and stronger measures than the ones that were eventually implemented on November 2[nd].

## Data and methods

### *Science Barometer* surveys

The *Science Barometer* is an annual population survey done by WiD. In 2020, two additional survey waves focused on public perceptions of science and research during the COVID-19 pandemic, supplementing the annual survey waves. All *Science Barometer* surveys rely on nationally representative samples of German-speaking residents aged 14 years and older who report their beliefs about science and research in computer-assisted telephone interviews (dual frame of landlines/mobile phones, 80:20). Sampling frames are provided by the Arbeitskreis Deutscher Marktforschungsinstitute (2020), which is the leading business association for private market and social research institutes in Germany.

In this contribution, we draw on data from four cross-sectional *Science Barometer* survey waves: the PreCOVID 09/19 wave conducted from 3–10 September 2019, the COVID 04/20 wave on 15–16 April 2020, the COVID 05/20 wave on 25–26 May 2020, and the COVID 11/20 wave from 3–9 November 2020. All waves were conducted by market research firm Kantar (www.kantardeutschland.de) as part of an omnibus survey on behalf of WiD. Informed consent was verbally obtained from all respondents before participation in the telephone interviews. The data sets analyzed in this article included 29 respondents aged 14 to 17 years. Consent from the parents or legal guardian of these respondents was not required in accordance with the German legislation and the requirements of the German Business Association for Market and Social Research (ADM). The PreCOVID 09/19 and the COVID 11/20 surveys were the regular annual survey waves (including further questions not covered in this paper), whereas the COVID 04/20 and the COVID 05/20 surveys were conducted additionally and included only a few questions. Altogether, the four surveys allowed us to track people's trust in science and research, their perspective on science–politics relationships, and the impact of explanatory factors thereof.

English translations of the full questionnaires are available at the Open Science Framework (OSF): https://osf.io/czn4g/. At the beginning of all science-related questions, the interviewers stated that social science and humanities where included when they referred to science and research. In the COVID 04/20 and COVID 05/20 surveys, respondents were asked to assess how much they trust in science and research directly after this statement. Subsequently, the COVID-19 pandemic was introduced as reference frame for the following questions. For each variable selected for our analyses, respondents indicated their level of agreement or their level of trust (depending on the item) on a verbalized 5-point Likert-type scale (trust completely/completely agree, trust somewhat/somewhat agree, undecided, distrust somewhat/somewhat disagree, distrust completely/completely disagree).

### Data

*PreCOVID 09/19.* A total of $N = 1,017$ respondents completed the interviews (51% female, $M_{age} = 48.6$, $SD_{age} = 19.8$). Thirty four percent had a low level, 30% had a medium level, and 31% had a high level of formal education; 4% of respondents were school students.

*COVID 04/20.* A total of $N = 1,009$ respondents completed the interviews (51% female, $M_{age} = 48.6$, $SD_{age} = 20.1$). Thirty four percent had a low level, 30% had medium level, and 31% had a high level of formal education; 4% of respondents were school students.

*COVID 05/20.* A total of $N$ = 1,021 respondents completed the interviews (51% female, $M_{age}$ = 48.9, $SD_{age}$ = 19.6). Of these, 34% had a low level, 30% had medium level, and 32% had a high level of formal education; 4% of respondents were school students.

*COVID 11/20.* A total of $N$ = 1,016 respondents completed the interviews (51% female, $M_{age}$ = 49.0, $SD_{age}$ = 19.7). Here, 34% had a low level, 30% had medium level, and 32% had a high level of formal education; 4% of respondents were school students.

Note that we excluded school students from all analyses due to their small absolute numbers within the samples.

## Variables and analyses

A detailed introduction of all variables is provided within the Results Sections where appropriate. The variable *trust in science and research* and further items measuring science-related beliefs are listed in Table 1, which also includes descriptive statistics and the survey in which they were included. Demographic variables (age, gender, education, whether children younger than 14 years were in the respondents' household, and political orientation) were measured in all waves, (see online supplement, S1 Table).

All analyses relied on weighted survey data, which were handled with the R package survey v4.0 [34]. To obtain results representative of the German population, data were weighted regarding landline/mobile probability, gender, age, occupation, education, federal state as well as size of town and household. The analyses could be reproduced with the R syntax, which we share publicly with the complete survey data and original questionnaires at https://osf.io/czn4g/.

Several belief variables exhibited serious skewness. We report means (*M*) and standard deviations (*SD*) for the trust item and for all further science-related beliefs, but these parameters must therefore be considered with care (Table 1). To compare item means between surveys, we used Mann-Whitney U-tests in our first set of RQ 1 analyses as they are rather robust to skewness (S2 Table). To answering RQ 1 we also reported the composition of different responses by presenting relative response frequencies (100% = all valid responses), contrasting the top-two levels of agreement (i.e., collapsing agreement (= 4) and complete agreement (= 5)) with the bottom-two levels of agreement (i.e., collapsing disagreement (= 2) and complete disagreement (= 1)). Additionally, when the proportion of respondents claiming to be undecided about a particular issue was remarkably large, we report that as well (S1–S27 Figs provide the proportions of responses for all belief items).

For a first overview of the relationship between trust in science and the further variables we provide zero-order correlations (S3 Table). To examine the explanatory power of demographic characteristics and science-related beliefs for *trust in science and research* (RQ 2), we then ran a series of survey-weighted multiple regression analyses. In a first step, we fitted four separate regression models, (one for each survey wave). Each of these included the same set of demographic characteristics as predictor variables with *trust in science and research* as the outcome variable (Table 2). In a second step, we fitted two extended regression models, one for COVID 04/20 and one for COVID 11/20. Each of these included the same set of demographic characteristics and the science-related beliefs that were measured in *both* COVID 04/20 *and* COVID 11/20 as predictor variables, with *trust in science and research* as the outcome variable (Table 3). In a third step, we fitted two additional extended regressions models, one for COVID 04/20 and one for COVID 11/20. Each of these included the same set of predictor variables supplemented by science-related beliefs that were exclusively measured *in either* COVID 04/20 or COVID 11/20, with *trust in science and research* as the outcome variable (S6 Table).

To examine temporal variability in the capacity of our predictor variables to explain trust in science (RQ 2), we ran further regression analyses (henceforth: interaction tests). Each of

**Table 1. Descriptive statistics of variables under scope.**

| | 09/2019 | | 04/2020 | | 05/2020 | | 11/2020 | |
|---|---|---|---|---|---|---|---|---|
| | M (SD) | Skew-ness | M (SD) | Skew-ness | M (SD) | Skew-ness | M (SD) | Skew-ness |
| Trust in science and research | 3.44 (0.85) | -0.36 | 4.00 (0.96) | -0.86 | 3.87 (0.99) | -0.60 | 3.71 (0.96) | -0.60 |
| Political decisions should be based on scientific evidence.[a] | 3.65 (0.97) | -0.36 | 4.26 (0.88) | -1.19 | 4.03 (1.24) | -1.15 | 4.24 (0.94) | -1.19 |
| It is not up to scientists to get involved in politics. | 2.66 (1.37) | 0.31 | 2.91 (1.35) | 0.05 | | | 3.20 (1.36) | -0.19 |
| Trust in statements on Corona made by politicians (09/2019: Trust in politics) | 2.52 (1.02) | 0.04 | 3.24 (1.09) | -0.44 | | | 2.86 (1.17) | -0.16 |
| Trust in statements on Corona made by journalists (09/2019: Trust in media) | 2.66 (0.98) | -0.13 | 2.88 (1.08) | -0.13 | | | 2.72 (1.07) | 0.01 |
| Trust in statements on Corona made by family members, friends and acquaintances | | | 3.10 (1.13) | 0.02 | | | 2.87 (1.09) | 0.05 |
| Reasons to trust: Because scientists are experts in their field. | 3.87 (0.91) | -0.55 | | | | | 3.97 (1.03) | -0.98 |
| Reasons to trust: Because scientists work according to rules and standards procedures. | 3.53 (1.07) | -0.47 | | | | | 3.85 (0.99) | -0.58 |
| Reasons to trust: Because scientists do research in the public interest. | 3.30 (1.04) | -0.25 | | | | | 3.43 (1.02) | -0.18 |
| Reasons to distrust: Because scientists often make mistakes. | 2.83 (1.11) | 0.19 | | | | | 2.63 (0.98) | 0.17 |
| Reasons to distrust: Because scientists often adjust results to their own expectations. | 3.24 (1.04) | -0.16 | | | | | 2.92 (1.03) | -0.05 |
| Reasons to distrust: Because scientists are strongly dependent on the funders of their research. | 3.87 (0.97) | -0.72 | | | | | 3.50 (1.19) | -0.43 |
| Controversies between scientists regarding corona are helpful . . . . . ... | 3.92 (0.93) | -0.45 | 3.98 (0.98) | -0.78 | | | 3.89 (1.06) | -0.92 |
| Most scientists currently speaking up differentiate clearly between what they know for sure and what are open . . .. | | | 3.66 (1.02) | -0.25 | | | 3.45 (0.98) | -0.38 |
| Science and research on Corona are so complicated that I do not understand much of it. | 3.12 (1.14) | -0.05 | 2.85 (1.26) | 0.07 | | | 3.01 (1.25) | 0.04 |
| Beliefs in the promise of science and research on Covid-19[b] | | | 3.75 (0.66) | -0.32 | | | | |
| Skeptical attitudes regarding the pandemic[c] | | | | | | | 2.65 (0.96) | 0.26 |
| We should rely more on common sense when dealing with corona and we do not need any scientific studies. . . .. | 3.02 (1.24) | -0.03 | 2.26 (1.42) | 0.71 | | | 2.34 (1.42) | 0.64 |
| I think the current measures against Corona are appropriate. | | | 4.07 (1.12) | -1.12 | | | 3.78 (1.29) | -0.83 |
| Subjective probability to get infected and expected severity of infection [d] | | | | | | | 2.81 (0.90) | 0.13 |
| The knowledge of scientists is important to slow the spreading of the coronavirus in Germany" | | | 4.50 (0.87), | | | | | |
| In the foreseeable future, science and research will provide vaccines or medication that will allow us to successfully deal with Corona | | | 3.71 (1.22) | | | | | |
| Science and research do not properly understand the coronavirus yet" (M = 3.00, SD = 1.31). | | | 3.00 (1.31) | | | | | |
| Scientists do not tell us everything they know about the coronavirus | | | | | | | 3.06 (1.33) | |
| It is important to also get information on the coronavirus from outside the scientific community | | | | | | | 3.10 (1.38) | |
| The coronavirus pandemic is being made into a bigger deal than it actually is | | | | | | | 2.56 (1.47) | |

*(Continued)*

**Table 1.** (Continued)

| | 09/2019 | | 04/2020 | | 05/2020 | | 11/2020 | |
|---|---|---|---|---|---|---|---|---|
| | M (SD) | Skew-ness | M (SD) | Skew-ness | M (SD) | Skew-ness | M (SD) | Skew-ness |
| There is no real proof that the coronavirus really exists | | | | | | | 1.84 (1.36) | |
| Subjective probability to get infected | | | | | | | 2.67 (1.13 | |
| Expected severity of the illness in case of an infection" | | | | | | | 2.94 (1.24) | |

*Note*. Descriptive statistics were computed using survey weights (R package survey v4.0; [34]).

[a] In the 09/19 wave, this item was introduced in a reference to climate change research and policy-making; in the 04/20, 05/20 and 11/20 waves it was introduced in a reference to the Covid-19 pandemic.

[b] Beliefs in the promise of science and research on Covid represents the mean value across the three survey items: "The knowledge of scientists is important to slow the spreading of the coronavirus in Germany", "In the foreseeable future, science and research will provide vaccines or medication that will allow us to successfully deal with Corona", and "Science and research do not properly understand the coronavirus yet" (inverted).

[c] Skeptical attitudes regarding the pandemic = mean score of survey items represents the mean value across the four survey items "Scientists do not tell us everything they know about the coronavirus", "It is important to also get information on the coronavirus from outside the scientific community", "The coronavirus pandemic is being made into a bigger deal than it actually is", "There is no real proof that the coronavirus really exists".

[d] "Subjective probability to get infected and expected severity of infection" represents the mean of "Subjective probability to get infected" and "Expected severity of the illness in case of an infection" (scale: Most likely, rather likely, maybe, rather unlikely, most unlikely).

which relied on the combined data of two survey waves. For example, to test differences in the explanatory power of the predictors between PreCOVID 09/19 and COVID 04/20, we merged the datasets of these two surveys, added a dummy variable indicating time of data collection (0 = 09/19; 1 = 04/20), defined *trust in science and research* as the outcome variable, and specified as predictors the dummy variable, the demographic (and belief) variables, and the interactions of the dummy and demographic (and belief) variables. Demographic (and belief) variables of interaction terms that reached significance could then be assumed to have changed in their capacity to explain trust in science between the two respective surveys (S4 and S5 Tables).

Because the assumption of normality of residuals was violated for all regression models, we employed bootstrapping [35,36] to estimate standard errors and confidence interval bounds of regression coefficients, using the bias-corrected and accelerated method (BC$_a$; [37]). Results of bootstrapped and original estimates only differed marginally (which can be retraced by running the R syntax we shared at OSF, see above).

## Results

### Part 1: Examining trust in science and further science-related beliefs before and during the pandemic (RQ 1)

First, we present the findings for the item *trust in science and research* and then continue with the items regarding further science-related beliefs. All differences were tested with U-tests, all statistics for U-tests are available in the online supplement, S2 Table.

**Trust in science and research.** The mean response to the item *How much do you trust in science and research* increased after the outbreak of the pandemic in the first COVID 04/19 survey in April 2020. This remained nearly at this level in May (COVID 05/20) and decreased in November (COVID 11/20). The increases between the PreCOVID 09/19 and the within-pandemic trust means are statistically significant. After April 2020, a (comparatively smaller)

**Table 2. Predicting trust in science and research before and during the Covid-19 pandemic.**

| | 09/2019 | | | | 04/2020 | | | |
|---|---|---|---|---|---|---|---|---|
| | b | p | 95% CI | SE | b | p | 95% CI | SE |
| Intercept | **2.86** | **<0.001** | [2.50, 3.21] | 0.18 | **2.63** | **<0.001** | [1.933, 3.27] | 0.34 |
| Gender (1 = female) | -0.16 | 0.062 | [-0.34, 0.00] | 0.09 | -0.06 | 0.556 | [-0.27, 0.15] | 0.11 |
| Age (1 = 60 years or older) | -0.11 | 0.258 | [-0.28, 0.11] | 0.10 | -0.15 | 0.228 | [-0.36, 0.11] | 0.12 |
| Education (1 = A-level) | **0.24** | **0.008** | [0.06, 0.42] | 0.09 | **0.29** | **0.009** | [0.08, 0.51] | 0.11 |
| Children aged < 14 years in household (1 = yes) | 0.00 | 0.968 | [-0.19, 0.23] | 0.11 | -0.17 | 0.219 | [-0.45, 0.08] | 0.14 |
| Populist party preference (1 = AfD) | 0.11 | 0.353 | [-0.13, 0.34] | 0.12 | -0.57 | 0.057 | [-1.17, 0.04] | 0.30 |
| Political decisions should be based on scientific evidence.[a] | **0.17** | **<0.001** | [0.07, 0.25] | 0.05 | **0.34** | **<0.001** | [0.21, 0.47] | 0.07 |
| Adj. $R^2$ | .09 | | | | .15 | | | |
| F value | $F(6, 894) = 3.85, p < .001$ | | | | $F(6, 943) = 10.32, p < .001$ | | | |
| N | 901 | | | | 928 | | | |
| | 05/2020 | | | | 11/2020 | | | |
| | b | p | 95% CI | SE | b | p | 95% CI | SE |
| Intercept | **2.65** | **<0.001** | [2.24, 3.24] | 0.25 | **2.43** | **<0.001** | [2.05, 2.93] | 0.22 |
| Gender (1 = female) | -0.20 | 0.069 | [-0.43, 0.00] | 0.11 | -0.17 | 0.060 | [-0.35, 0.00] | 0.09 |
| Age (1 = 60 years or older) | -0.13 | 0.218 | [-0.35, 0.07] | 0.11 | -0.02 | 0.839 | [-0.22, 0.16] | 0.09 |
| Education (1 = A-level) | **0.25** | **0.019** | [0.03, 0.46] | 0.11 | **0.55** | **<0.001** | [0.39, 0.74] | 0.09 |
| Children aged < 14 years in household (1 = yes) | -0.02 | 0.928 | [-0.33, 0.47] | 0.21 | -0.22 | 0.133 | [-0.55, 0.02] | 0.14 |
| Populist party preference (1 = AfD) | -0.34 | 0.307 | [-1.23, 0.19] | 0.34 | -0.47 | **0.045** | [-0.93, 0.01] | 0.23 |
| Political decisions should be based on scientific evidence.[a] | **0.32** | **<0.001** | [0.20, 0.42] | 0.06 | **0.29** | **<0.001** | [0.18, 0.38] | 0.05 |
| Adj. $R^2$ | .22 | | | | .24 | | | |
| F value | $F(6, 943) = 12.57, p < .001$ | | | | $F(6, 923) = 26.39, p < .001$ | | | |
| N | 950 | | | | 930 | | | |

*Note*. Analyses used survey weights and were computed using the R package survey v4.0 [34]. In all regression models, the assumption of normality of the residuals was violated (which can be retraced by running the R syntax we share, see Methods section); therefore, standard errors and confidence interval bounds (95%, two-sided) of $b$ coefficients were bootstrapped. Bootstrapping was done with the R package boot v1.3–25 [35] using the bias-corrected and accelerated method (BC$_a$; [37]), which accounts for the skewness and lack of symmetry in the observed data [36]. Boldface = p < .05.

[a] In the 09/19 wave, this item was introduced in a reference to climate change research and policy-making; in the 04/20, 05/20 and 11/20 waves it was introduced in a reference to the COVID-19 pandemic.

decrease started, which was beyond random fluctuations after approximately 6 months: In November 2020, trust was significantly lower in contrast to the April and May surveys.

The large increase in trust after the beginning of the pandemic is also visible in the proportions of those who trust in science and research somewhat or completely. Since trust in science and research has first been measured in the *Science Barometer* in 2017, the proportion of Germans who claimed to have trust in science and research has been around 50%; in September 2019 it was 46%. However, after the outbreak of the pandemic in April 2020, and thus just half year later, 73% of the respondents indicated trust in science and research. This shift resulted from a drop in the proportion of respondents who claimed to be *undecided* (09/19: 46%; 04/20: 21%; 05/20: 24%; 11/20: 31%), while the proportion of those who claimed to distrust science and research somewhat remained stable during the COVID-19 pandemic (09/19: 8%; 04/20: 6%; 05/20: 9%; 11/20: 7%).

**Endorsement of science-based politics, trust in politicians and trust in further actors.** Another aspect of public's trust in science is reflected by the item "*Political decisions (regarding Corona) should be based on science*", presented in all four surveys. Similar to trust in science, the overall endorsement of science-based politics has been higher in the context of the

**Table 3. Predicting trust in science and research with additionally focusing on beliefs about science and the pandemic.**

| | Trust in science | | | | | | | |
|---|---|---|---|---|---|---|---|---|
| | 04/2020 | | | | 11/2020 | | | |
| | *b* | *p* | 95% CI | *SE* | *b* | *p* | 95% CI | *SE* |
| Intercept | **2.20** | **<0.001** | [1.25, 3.19] | 0.49 | **2.50** | **<0.001** | [1.84, 3.26] | 0.36 |
| Gender (1 = female) | -0.10 | 0.300 | [-0.26, 0.10] | 0.09 | -0.24 | **0.005** | [-0.43, -0.09] | 0.09 |
| Age (1 = 60 years or older) | -0.19 | 0.087 | [-0.41, 0.04] | 0.11 | -0.08 | 0.405 | [-0.26, 0.11] | 0.09 |
| Education (1 = A-level) | 0.14 | 0.135 | [-0.04, 0.32] | 0.09 | **0.34** | **<0.001** | [0.18, 0.50] | 0.08 |
| Children aged < 14 years in household (1 = yes) | -0.15 | 0.245 | [-0.42, 0.09] | 0.13 | -0.15 | 0.207 | [-0.38, 0.09] | 0.12 |
| Populist party preference (1 = AfD) | -0.27 | 0.397 | [-0.92, 0.36] | 0.32 | -0.11 | 0.671 | [-0.58, 0.49] | 0.26 |
| Political decisions should be based on scientific evidence. [a] | 0.13 | 0.048 | [-0.01, 0.24] | 0.06 | **0.11** | **0.045** | [0.00, 0.22] | 0.06 |
| It is not up to scientists to get involved in politics. | -0.01 | 0.799 | [-0.08, 0.07] | 0.04 | 0.03 | 0.408 | [-0.03, 0.09] | 0.03 |
| Trust in statements on Corona made by politicians (09/19: Trust in politics) | 0.11 | 0.101 | [-0.02, 0.25] | 0.07 | 0.07 | 0.249 | [-0.05, 0.19] | 0.06 |
| Trust in statements on Corona made by journalists (09/19: Trust in media) | **0.22** | **<0.001** | [0.11, 0.35] | 0.06 | **0.17** | **0.001** | [0.06, 0.28] | 0.05 |
| Trust in statements on Corona made by family members, friends and acquaintances | -0.06 | 0.328 | [-0.18, 0.05] | 0.06 | -0.05 | 0.334 | [-0.16, 0.04] | 0.05 |
| Controversies between scientists regarding corona are helpful because they help to ensure. . . . . . | **0.22** | **<0.001** | [0.11, 0.34] | 0.06 | **0.12** | **0.017** | [0.03, 0.24] | 0.05 |
| Most scientists currently speaking up differentiate clearly between what they know for sure. . . .. | -0.05 | 0.383 | [-0.15, 0.06] | 0.05 | -0.01 | 0.815 | [-0.12, 0.07] | 0.05 |
| Science and research on Corona are so complicated that I do not understand much of it. | -0.05 | 0.238 | [-0.14, 0.03] | 0.04 | **-0.09** | **0.010** | [-0.17, -0.02] | 0.04 |
| We should rely more on common sense when dealing with corona and we do not need any scientific . . . .. | **-0.10** | **0.034** | [-0.18, -0.01] | 0.05 | -0.08 | 0.071 | [-0.16, 0.01] | 0.04 |
| I think the current measures against Corona are appropriate. | 0.06 | 0.234 | [-0.03, 0.16] | 0.05 | 0.05 | 0.248 | [-0.03, 0.15] | 0.04 |
| *Adj. R²* | 0.33 | | | | 0.34 | | | |
| *F value* | $F(15, 816) = 13.56, p < .001$ | | | | $F(15, 849) = 14.84, p < .001$ | | | |
| *N* | 832 | | | | 865 | | | |

*Note.* Analyses used survey weights and were computed using the R package survey v4.0 [34]. In all regression models, the assumption of normality of the residuals was violated (which can be retraced by running the R syntax we share, see Methods section); therefore, standard errors and confidence interval bounds (95%, two-sided) of *b* coefficients were bootstrapped. Bootstrapping was done with the R package boot v1.3–25 [35] (Ripley, 2020) using the bias-corrected and accelerated method (BC$_a$; [37], which accounts for the skewness and lack of symmetry in the observed data [36]. Boldface = p < .05.

[a] In the 09/19 wave, this item was introduced in a reference to climate change research and policy-making; in the 04/20, 05/20 and 11/20 waves it was introduced in a reference to the COVID-19 pandemic.

COVID-19 pandemic compared to the PreCOVID 09/19 survey, when these items were put in the context of climate change regarding the Fridays for Future protests. Agreement with this statement was 55.6% (top two) in 09/19 compared to 81% in 04/20. This shift was again primarily due to a drop in the proportion of *undecided respondents* (09/19: 35%; 04/20: 16%), while the proportion of disagreement was consistently small. However, compared to trust in science, there was no significant decrease on this item over the course of the pandemic.

With regard to public' endorsement of science-based politics, it is also interesting to look at the results for agreement with the item "*It is not up to scientists to get involved in politics*". The percentage of respondents who disapproved of scientists being actively involved in politics (i.e., agreement with this item) was slightly below the scale midpoint before the pandemic. However, this increased significantly when the pandemic emerged and during the pandemic. In contrast to the previous question about science-based politics, the German people were more divided about this issue. The levels of agreement were 29% (top two) in 09/19, 34% in 04/20 and 44% in 11/20, while disagreement was 50% (bottom two) in 09/19, 39% in 04/20 and

31% in 11/20. Thus, although the percentage of respondents who disapproved of scientists engaging in politics was not very high, but it clearly increased during the pandemic.

Respondents were also asked to assess their trust of statements on COVID-19 made by different actors. We specifically focused on their trust assessments of *politicians*, *journalists*, and *family members*, *friends*, *and acquaintances*. Note that the PreCOVID responses of 09/19 were based on items for *trust in politics* (not *statements by politicians* on COVID-19) and *trust in media* (not *statements by journalists* on COVID-19); thus, they are not literately equivalent to the items used in the COVID-19 surveys. Trust in the latter two groups started below the midpoint of the scale, increased in the context of the outbreak of the pandemic, and then decreased from April to November 2020.

Trust in *politicians* was significantly higher in 4/20 after the outbreak of the pandemic and then decreased, but was in 11/20 still higher than before the pandemic (two agreement 09/19: 16%; 04/20: 44%; 11/20: 32%). In contrast, trust in *journalists* in 11/20 fell back to a level that is statistically not different from the PreCOVID survey in 09/19. The proportion of top two agreement levels was 18% in 09/19, 27% in 04/20, and 22% in 11/20, while disagreement was 39% in 09/19, 31% in 04/20 and 39% in 11/20. In all three surveys measuring trust in *journalists or the media*, large groups of respondents indicated they were undecided (09/19: 43%; 04/20: 42%; 11/20: 39%).

At the beginning of the pandemic, *family members*, *friends and acquaintances* were clearly considered more trustworthy than *politicians* and *journalists* regarding statements related to COVID-19. However, this assessment decreased slightly, reaching a similar level as trust in politicians in November 2020. However, the group of respondents indicating they were undecided about the trustworthiness of *family members*, *friends*, *and acquaintances* remained large (04/20: 41%; 11/20: 41%; top two agreement 04/20: 32%; 11/20: 25%, bottom two disagreement 04/20: 27%, 11/20: 34%).

**Reasons for (dis)trust along the dimensions of expertise, integrity, and benevolence.** The surveys in 09/19 and 11/20 asked respondents about their levels of agreement with different reasons for trusting and distrusting scientists. The reasons for trust were as follows: *because scientists are experts in their field* (expertise), *because scientists work according to rules and standards* (integrity), and *because scientists do research in the public interest* (benevolence). The reasons for distrust were as follows: *because scientists often make mistakes* (expertise), *because scientists often adjust results to their own expectations* (integrity) and *because scientists are strongly dependent on the funders of their research* (benevolence) [38,39]. The reason *expertise* was most pronounced in September 2019 and November 2020, followed by *integrity* and finally *benevolence* reasons. The proportion of agreement (top two) for the reason *expertise* was 67% in 09/19 and 71% in 11/20, while disagreement (bottom two) was low, with 6% in 09/19 and 7% in 11/20. A similar pattern emerged for the *integrity* assessments: agreement with the reason *integrity* was 55% in 09/19 and 64% in 11/20, while disagreement (bottom two) was larger before the pandemic (17% in 09/19) compared to during the pandemic (7% in 11/20). Regarding *benevolence* as a reason for trust, the level of agreement was 42% in 09/19 and 45% in 11/20, while disagreement (bottom two) was 19% in 09/19 and 15% in 11/20 during the pandemic. The proportion of undecided respondents was relatively large: 09/19: 39%, 11/20: 41%. Agreement with *expertise* and *integrity* as reasons for trusting scientists also increased significantly from the 09/19 to the 11/20 survey, while the increase in agreement with the benevolence item was not significant.

The results for respondents who agreed with the reasons for distrust follow an inverse pattern. The most pronounced reason for distrust was the perceived violation of *benevolence*, followed by violations of *integrity* and then *expertise*. The agreement with *benevolence* violations as a reason for distrusting scientists was 65% in 09/19 and 52% in 11/20, whereas the levels of

disagreement were 12% in 09/19 and 19% in 11/20. Moreover, agreement with the *integrity* violations was 40% in 09/19 and 26% in 11/20, whereas levels of disagreement were 22% in 09/19 and 31% in 11/20. Finally, agreement with the *expertise* violations was 20% in 09/19 and 17% in 11/20, whereas levels of disagreement were 35% in 09/19 and 44% in 11/20. The agreement with each of these reasons for distrust decreased from the 09/19 survey to the 11/20 survey.

**Uncertainty, disagreements among scientists, and the challenge for laypersons determining who is right and who is wrong.** As the SARS-CoV-2 virus represents a new phenomenon for research, this necessarily implies that there will be discrepant voices among scientists researching it. Since 2018, the annual *Science Barometers* have included an item about public's perceptions of disagreement among scientists. Levels of agreement for "*Controversies between scientists regarding Corona are helpful because they help to ensure that the right research results prevail*" were available for the 04/20 and 11/20 surveys. (In the 09/19 survey, it was without the reference to a specific topic). The proportions of responses show that about two-thirds of respondents endorsed this position (top two) and only a small proportion of respondents disagreed (bottom two: 09/19: 5%; 04/20: 5%; 11/20: 10%). The scores reveal overall positive and stable views on disagreements between scientists without any significant differences across the surveys.

Another aspect of scientific uncertainty has only been covered in surveys during the pandemic: "*Most scientists currently speaking up differentiate clearly between what they know for sure and what are open questions on Corona*". Overall, about half of the respondents agreed with this statement; (top two agreement 04/20: 54%, 11/20: 47%, bottom two disagreement 04/20: 11%, 11/20: 12%). However, agreement with this statement declined between 04/20 and 11/20.

A similar pattern occured when investigating personal uncertainty caused by perceived difficulty to understand science, which was measured with the item "*Science and research (on Corona) are so complicated that I do not understand much of it*". It should be noted that, this was asked in 09/9 without reference to a specific topic. The respondents were nearly evenly divided regarding their feeling of understanding science in 09/19. The proportion of participants who agreed with this statement was 35% (top two) in 09/19, 31% in 04/20 and 35% in 11/20, while the level of disagreement was 28% (bottom two) in 09/19, 39% in 04/20 and 36% in 11/20. The level of respondents who felt that they did not understand science and research declined after the beginning of the pandemic.

**Potency beliefs and wariness beliefs about science and personal stances toward the pandemic.** Within the additional 04/20 survey wave and the annual 11/20 survey wave of the *Science Barometer*, two additional sets of items captured beliefs about the potency of science for coping with COVID-19. Confidence in the potency of science was measured in the 04/20 survey with three survey items "*The knowledge of scientists is important to slow the spreading of the coronavirus in Germany*", (top two agreement of 89%, bottom two disagreement 5%); "*In the foreseeable future, science and research will provide vaccines or medication that will allow us to successfully deal with Corona*", (top two agreement of 61%, bottom two disagreement of 19%); "*Science and research do not properly understand the coronavirus yet*", (top two agreement of 38%, bottom two disagreement of 37%) which is the inverse version of a hopeful belief. Agreement with these items reflected the magnitude of positive expectations about the potency of science.

In 11/20, wariness beliefs were measured by four items "*Scientists do not tell us everything they know about the coronavirus*", (top two agreement of 41%, bottom two disagreement of 35%); "*It is important to also get information on the coronavirus from outside the scientific community*"; (top two agreement of 40%, bottom two disagreement of 32%); "*The coronavirus*

*pandemic is being made into a bigger deal than it actually is"*; (top two agreement of 30%; bottom two disagreement of 53%); "*There is no real proof that the coronavirus actually exists"*; (top two agreement of 16%, bottom two disagreement of 75%). The PreCOVID 09/19 survey and also the during-pandemic surveys included one further item which corresponds with a measure of wariness beliefs: '*We should rely more on common sense (when dealing with Corona) and we do not need any scientific studies for this'* (top two agreement 09/19: 35%, 04/20: 23%, 11/20: 24%, bottom two disagreement 09/19: 33%, 04/20: 62%, 11/20: 61%). The average agreement with this statement decreased after the pandemic began. Corresponding with the results of the previous block of potency-related items, wariness beliefs were endorsed to a smaller degree after the pandemic emerged, and, even more interestingly, they declined during the pandemic.

The item "*I think the current measures against Corona are appropriate"* was part of the 04/20 and 11/20 survey waves. Although the average acceptance of the current measures was relatively high, it declined notably. The 11/20 survey included some further items about respondents' personal stance toward the pandemic: *subjective probability to get infected*, and *expected severity of the illness in case of an infection*. Both of these were included in the analyses regarding RQ 2, presented below.

## Part 2: What are the explanatory factors of public's trust in science and its variability?

A further goal of this contribution is to explore the factors that may affect trust in science and, hence, contributing to explaining the variability in the public's trust. All multiple regression models considered the following demographic information:

*Age*: (<60/≥60 years). Age is associated with increased lethality of COVID-19 and with a higher risk of suffering a severe form. Accordingly, it might be possible that respondents who are 60 years old or above might endorse different stances toward science in the context of COVID-19 compared to those who are younger.

*Gender*: (male/female). Studies suggest gender-related differences in the prevalence of severe forms of the disease, in the perceived seriousness of the COVID-19 pandemic, and the acceptance of non-pharmaceutical interventions (NPIs) [40,41].

*Education*: (non-academic/academic). We collapsed participants with low and medium levels of formal education versus those with at least the *Abitur*, the final exam after 12 or 13 years of school education, which is the mandatory requirement for studying at a German university.

*Children in household*: (children younger than 14 years in the respondents' household, yes/no). This factor was considered relevant, since in 2020 the closure of child-care facilities and schools was an important element of the lockdown measures in Germany. Therefore, this variable was used as a proxy for potential pandemic-related burdens.

*Populist political orientation*: (AfD supporters, yes/no).The *Science Barometer* asks respondents for their preferences among the political parties in Germany. We coded AfD (the populist right-wing party "Alternative for Germany") versus non-AfD supporters as a binary indication of populist attitudes [42]. Generally, populist ideas do not only criticize a political establishment, but they may also challenge academic elites, suggesting that scientific knowledge (e.g., about COVID-19) is useless, ideologically biased, and inferior to ordinary people's common sense [43]. During the COVID-19 pandemic, public opposition against the German government's containment measures was promoted by right-wing political movements, (especially when suggesting that COVID-19 is no threat for public health).

**General predictors for public trust in science before and during the pandemic.** Zero-order correlations provide a first overview about the relationship between *trust in science and*

*research* and each demographic and belief variable (S3 Table). However, they do not control for relationships among these variables. We conducted separate regression analyses for all survey waves, using the same predictor variables (*age*, *gender*, *education*, *children in household*, *populist party preference*, and *respondents' endorsement of science-based politics*). For each survey, the regression model showed significance and displayed a consistent pattern of significant predictors (see Table 2). Even though the same predictors were significant, the amount of explained variance increased from the PreCOVID 09/19 survey ($R^2$ = .09) to the COVID 11/20 survey ($R^2$ = .24) (Table 2). It appeared that the predictors gained in importance for individuals' trust in science during the pandemic. *Education* showed the highest correlation with trust in science in the COVID 11/20 survey, when people in Germany had experienced the pandemic for approximately ten months. Meanwhile, *endorsement of science-based politics* was most strongly associated with trust in April and May 2020, (in the first months after the outbreak of the pandemic in Germany).

In addition, we ran interaction tests to examine the temporal variability of these predictors' explanatory power throughout the four survey waves (see S4 Table). The education x time interaction reached significance when comparing the Pre-COVID 09/19 and COVID 11/20 surveys (*b* = 0.32, SE = 0.13, p < .05). This suggests that the effect of *education* on trust in science increased significantly between September 2019 and November 2020. However, when comparing shorter time spans (e.g., September 2019 and April 2020), we did not observe significant variability in the predictive power of *education*. Meanwhile, the association between *trust in science* and the *endorsement of science-based politics* increased significantly between September 2019 and April 2020 (b = 0.17, SE = 0.08, p < .05). Interestingly, the influence of *populist party preference* on trust in science also changed significantly over time: AfD voting was significantly more likely to reduce trust in April 2020 than in September 2019, (b = -0.68, SE = 0.32, p < .05). Moreover, *populist party preference* was more likely to predict lower trust levels in November 2020 compared to September 2019 (b = -0.58, SE = 0.26, p < .05).

**Science-related beliefs as predictors for trust in science during the pandemic in April and November 2020.** Both the 04/20 and 11/20 surveys contained a common set of additional variables measuring science-related beliefs. These belief variables captured the acceptance of measures taken and trust in statements on the COVID-19 pandemic by different actors (i.e., *politicians*, *journalists*, *family members*, *friends and acquaintances*). Hence, to explore factors shaping trust in science during the pandemic further, we extended the prior regression model and included these variables as predictors in addition to those predictors considered before (see above). Table 3 lists all predictors and reports regression coefficients, explained variance and test statistics considered in these analyses.

Both regression models (04/20 and 11/20) reached significance. The amount of explained variance was nearly equivalent with $R^2$ = .33, and $R^2$ = .34, respectively. In the following, we discuss the results for these additional belief variables that were included only in these two surveys.

Across both surveys *trust in statements on COVID-19 by journalists* was significantly related with trust in science and research. However, neither to *trust in statements from politicans* nor *trust in statements from family and friends*. Respondents' positive perception of *controversies among scientists as part of scientific progress* was predictive for trust in science in 04 /20 and in 11/20.

However, there were also differences between both surveys. Namely, respondents *advocating for common sense as basis of decisions* showed less trust in science in 04/20, a relationship not confirmed by data from 11/20. Moreover, the perception of *science on Corona to be too complicated* was negatively associated with trust in science in 11/20, but not in 04/20.

The interaction tests did not indicate significant temporal changes in the explanatory power of any of these predictors between 04/20 and 11/20 (S5 Table).

**A closer look: Considering additional beliefs, measured exclusively in 04/20 or 11/20.**
The 04/20 and 11/20 surveys included additional items that captured science-related beliefs
exclusively in either one of both surveys. These items also appear relevant in order to explain
public's trust in science and research. Therefore, we extended our exploration of potential
explanatory factors of trust in science and fitted regression models for each of these two sur-
veys. These models contained the additional items as predictors, besides the previously
reported ones, (see S6 Table for a summary of the results).

The COVID 04/20 survey additionally covered respondents' beliefs in the *promise of science*
with three items (see above Results Part 1), which we aggregated to a mean score (see S6 Table,
note a). Overall, the model yielded significance and explained 33% of the variance of trust in
science ($R^2 = .33$). The additional predictors capturing beliefs in the *promise of science* did not
significantly predict respondents' trust in science and research (while the effects of prior
inspected predictors still persisted).

The COVID 11/20 survey additionally measured *skeptical beliefs about science* (measured
with four items combined to a mean score) as well as *reasons for trusting and distrusting scien-
tists*, (each measured with three items, see above). Overall, the model reached significance and
explained 42% of the variance of trust in science ($R^2 = .42$). German people who endorsed
skeptical stances toward the COVID-19 pandemic and toward scientists (in the context of the
COVID-19 pandemic) showed less *trust in science and research* in general. Agreement with
*expertise* and *integrity* as reasons for trusting scientists was significantly related to trust in sci-
ence. In contrast, neither agreement with *benevolence* as a reason for trusting scientists nor
agreement with the three *reasons for distrusting* them was significantly related with trust in
science.

Furthermore, the COVID 11/20 survey included a set of items that scrutinized respondents'
stances toward the pandemic specifically, such as "The Corona pandemic is made into a bigger
deal than it actually is". These items referred to skeptical beliefs about the pandemic itself
rather than the scientific research about COVID-19. They were combined to a mean score (see
S6 Table), which was negatively related to trust in science (b = -.20; p < .05). In contrast, the
perceived risk to suffer from a COVID-19 infection was not related to trust in science.

## Discussion

### RQ 1: What are public perceptions of science, especially trust in science, before and during the pandemic in Germany, and how have they changed?

**Trust in science and research.**  There was a large increase in the public's trust in science
and research in Germany between September 2019 and April 2020 after the outbreak of the
pandemic. Further, there was a slight, but still remarkable, decline of trust in subsequent the
seven months, i.e., between April and November 2020. This increase in the proportion of
those who claimed to have trust in science and research was far beyond the typical fluctuation
of this score which has been found in previous years (50% +/- 4%) [44–46]. This high trust in
science was embedded in an even higher expectation of the potency of science for coping with
the pandemic. Nearly 90% of respondents agreed that scientific knowledge is important for
slowing the spread of the coronavirus.

However, this high expectation does not necessarily reflect a gullible faith in the potency of
science: For example, in April 2020, only 61% (not the 90% associated with the previously
mentioned question) of the German people were optimistic that science would be able to pro-
vide medication and vaccines to combat the pandemic.

The increase of trust in science was not only visible through responses to the single item
asking about trust in science. The PreCOVID 9/19 and the COVID 11/20 surveys asked about

*reasons for trust and distrust in scientists*. Respondents endorsed different reasons for trust and for distrust, and this pattern remained the same before and during the pandemic. However, it became more pronounced: The average endorsement of reasons for trust increased, while the average endorsement of reasons for distrust, decreased. This suggests that the increase of trust in science was not simply a shallow response to one rather general item.

**Science and politics.** The magnitude of trust in science and scientists as well as respondents' high expectations for science to solve problems need to be considered in a broader context that encompasses the role of politics as a crucial factor for effectively coping with the pandemic. Similar to trust in science, public endorsement of science-based politics increased significantly when the pandemic started. In April 2020, over 80% of respondents agreed that science should inform politics. This proportion was even larger than the proportion of respondents who claimed to have trust in science, and it did not decrease between April and November. Based on a representative online survey in Germany from April 2020, Post, Bienzeisler and Lohöfener [47] similarly reported very high agreement with the expectation that science should dominate policy-making. However, our findings do not suggest, that such strong endorsement of science-based politics necessarily implies a preference for mingling the differences between the roles of scientists and politicians. Moreover, we did not find an unequivocal support of an expertocracy (but see [47] for a different conclusion): when asked whether scientists *should not get involved in politics*, the respondents were divided. However, the proportion of those who thought this way increased after the outbreak of the pandemic. This aligns with our finding that trust in (statements by) *politicians*, which has typically been rather low in previous *Science Barometer* surveys, was above the midpoint of the scale after the outbreak by 04/20. After then it declined, but in 11/20 it was still higher than before the pandemic. With regard to trust in *journalists*, people seemed to realize that they depend on information from beyond their personal environment to cope with the pandemic. This conclusion is based on the increase (at the beginning of the pandemic) of trust in statements by *journalists* and the steady decrease of trust in information from *family members*, *friends*, *and acquaintances*.

**Uncertainty and the feeling of understanding.** In Germany, politicians and the media have referred to science and to scientists as the main *epistemic* authorities on the pandemic. Especially at the beginning of the pandemic, a few virologists and epidemiologists were very present in the media [23,47]. Basic facts from these fields were explained in popular TV and print formats [48]. Accordingly, in April 2020, at the beginning of the pandemic, fewer Germans had the impression that "*Science and research on Corona are too complicated to be understood by oneself*" than in previous surveys when participants were asked about their understanding of science and research in general. However, this elevated feeling of understanding melted away between April and November 2020. In the COVID 11/20 survey, the level of agreement with the "too complicated" statement was no longer significantly lower than in the PreCOVID 9/19 survey. This leads us to assume that the vast amount of information on virology and epidemiology available after the outbreak increased a shared feeling of understanding [49]; a widely used description to make fun of this shared (feeling of) understanding was "*Everyone is becoming a virologist just as everyone in Germany is a soccer expert*". Subsequently, when it became more obvious that (beyond the basic facts from virology) the pandemic was highly complicated and researchers faced numerous open questions, this shared feeling of understanding faded away. While this is only an assumption, it aligns with findings from the BfR Monitor (a weekly online survey), which asks how well informed Germans feel about COVID-19 issues. It shows that the proportion of respondents who felt very well or well informed decreased from 74% to 66% between April an November 2020 [19].

The scientific information as well as recommendations for prevention measures based on scientific explanations were not unequivocal. In addition, some media as well as some

politicians emphasized alleged controversies between scientists. For example, within the first few months after the outbreak, Germany's largest tabloid paper (Bild) has attacked the most prominent virologist (Prof. Drosten, see [48]) regarding a partly real, partly alleged controversy about scientific results that had been made public in a pre-print [50]. Against this background, it is remarkable that two-third of the respondents endorsed the claim that "*Controversies among scientists regarding the COVID-19 pandemic are helpful"*, a proportion similar to what has been found in pre-pandemic *Science Barometer* surveys. Overall, it is notable that most respondents—at least in April 2020—acknowledged that "*scientists clearly differentiate between what they know and what they do not (yet) know about the Covid pandemic"*.

**Distrust in science and denial of COVID-19.**   The opinion "*We should rely more on common sense and do not need science"* portrays a kind of rejection of science. Although the endorsement of this claim clearly decreased after the outbreak of the pandemic, the response pattern also points to a polarization of positions. Before the pandemic only 33% of respondents clearly rejected this claim with 23% undecided. After the outbreak, 62% rejected it and only 15% remained undecided; thus, about a quarter of Germans endorsed this kind of science rejection. As such, one intriguing question is whether this response pattern reliably estimates the proportion of those who are generally more skeptical of the scientific evidence about COVID-19. However, science skepticism and science denial are not coherent beliefs, meaning such an estimate depends strongly on the exact focus of the question. In November 2020, 16% of respondents agreed that *no scientific proof of the coronavirus exists*, and more than one-third of respondents endorsed a kind of suspiciousness with regard to the *openness of scientists* and about the *severity of the pandemic*. By contrast, only 5% of respondents rejected the claim that "*scientists are important for slowing the spread of the virus"*. This result is in accordance with our finding that a stable proportion of less than 10% of German people clearly disagree when asked about their *trust in science and research*. Overall, we conclude that a moderate version of science skepticism (the belief that one *should listen less to science in favor of everyday knowledge*) was much less endorsed after the outbreak of the pandemic. Nevertheless, one-quarter of Germans did not perceive science as the paramount epistemic tool for fighting the pandemic. The proportion of those who deliberately distrusted science was again much smaller, typically below 10%. However, when it comes to the COVID-19 pandemic some polarization does exist.

## RQ 2: What are the explanatory factors of public trust in science and its variability?

One striking finding was the stark increase of explained variance in trust in science from before to during the pandemic. This was evident even within the first set of our analyses which only included a few critical demographic variables and an *endorsement of science-based politics* item. This suggests that respondents' considerations about trust in science were more immersed in their personal reality, assuming that the demographic variables (such as education) actually mirrored this personal reality. The COVID-19 pandemic affected everybody's lives, rendering science, (presumably: health-related disciplines), more salient overall. When asked about trust in science, respondents probably interpreted the survey question more personally than respondents in previous surveys, and personal factors may have mattered more for their trust judgments. Our conclusion is illustrated by the variable *education*. While higher education was associated with trust in science in all surveys (see Table 2), this effect appeared strongest in 11/20. Above this, education also predicted the increase of trust between the Pre-COVID survey in 9/19 and the survey 14 months later (11/20).

This finding aligns with the above-discussed finding regarding respondents' decrease in their feeling of understanding, which negatively predicted trust in science. At the beginning of the

pandemic, most people felt that they had learned something about the virus and its effects. However, as the pandemic persisted, it became obvious that many issues around COVID-19 were rather complex, but having a better education could help respondents to cope with this complexity.

The nexus of trust in science and the endorsement of science-based politics intensified after the pandemic began. The increase of effect of *endorsement of science-based politics* on trust judgments between 09/19 and 04/20 supports our hypothesis that, in the context of the COVID-19 pandemic, trust in science is closely related to expectations about how politics should handle the pandemic.

This relationship was especially visible with regard to populist orientations, measured here as preference for the populist party AfD. After the pandemic began, the effect of populist party preference on trust in science increased significantly: In April 2020 and November 2020, AfD voters were more likely to have low trust in science than in September 2019. Accordinlgy, AfD preference went along with a weaker increase in trust (compared to the other voter with other party preferences) in the beginning and a stronger decrease of trust during the pandemic. This indicates that populist political orientation might fuel skeptical views on science as the pandemic unfolded [24,51]. This, in turn, corresponds with the fact that Germany's AfD aligned more and more with outright deniers of the pandemic–perhaps because it perceived recommendations by epidemiology and virology experts as incompetent, ideologically biased, and patronizing advice from the academic elite [43].

Correspondingly, the regression analysis of the 11/20 survey shows that *skeptical beliefs about the pandemic* were most predictive (negatively) of trust in science. These beliefs include the outright denial of the coronavirus and similar beliefs of wariness. In contrast, the rather neutral item asking for the *perceived appropriateness of the present measures against the pandemic* was not predictive of trust in science and neither was *trust in statements from politicans*. Both items ask for views on politics in broad sense, reflecting a quite general stance against politics and pandemic containment. (For example, people could be unhappy with the measures against the pandemic either because they are too weak or too strong). We assume that only specific political views (i.e. preference for populism) are predictive for trust in science, as suggested by our finding on the impact of populist orientation. This constitutes a conceptual and methodological challenge for future surveys (see below). However, it allows for the cautious conclusion that, at least until November 2020, the widely shared positive view on science was to some extent decoupled from the rapidly changing view on politics and its handling of the pandemic (with the exception of those who hold populist political positions).

**The sustainability of the present trust in science.** The most remarkable result of the surveys covered in this study was the huge increase of trust in science in the context of the outbreak of the pandemic. This increase was not trivial. In other social contexts and during previous epidemics, public trust in scientists was reduced [52]. However, German citizens' increase of trust in science surrounding the COVID-19 pandemic is in accordance with findings in other countries (Netherlands, [53]; USA, [29]; Switzerland, [54]; UK, [27]; international overview, [10,28,55]. The overall conclusion has been "Pandemic: Public feeling more positive about science" [55]. Importantly, most of these surveys were performed no later than autumn 2020. It is yet unclear how persistent the increased trust in science/scientists will be; for example, Battiston, Kashyap and Rotondi [2] have already described rapid decreases of public trust in science in Italy; Algan et al (based on a different survey) also reports decreases of trust in science in Italy, France and the United States [10].

The stark increase of public trust in science as well as the smaller decrease in this trust between April and November 2020 raise the question of sustainability. It has been argued that science was the straw that German people grasped at in despair when the worldwide crisis of COVID-19 appeared [56] (with regard to a US survey, see [57]). In contrast, we argue that the

increase of trust is more than a shallow, ephemeral development. The increase of trust in science was accompanied by similar changes in the acceptance of further science-related beliefs: respondents also agreed more with the idea that politics should rely on scientific evidence and that the power of (medical) science should be relied on for coping with COVID-19. Moreover, in spite of some turmoil in the (social) media about the variability of COVID-19 related findings, the belief that scientific controversies are fruitful for the progress of science was not endorsed less than before the pandemic. When asked for reasons for trust, beliefs in scientists' expertise as well as integrity (i.e. working according to rules and standard procedures) were predictive for trust (S6 Table)

Insofar, it can be assumed that the responses with regard to the trust item were not shallow surface responses. Rather they were embedded in respondents' increasing awareness of the role of science in fighting a pandemic. Accordingly, we argue that for the general public in Germany, science is not a straw to grasp at in despair. Rather it is an anchor in troubled times, even when trust in the politics of coping with the pandemic is decreasing.

However, we must also consider the gradual decrease of trust in science between April and November 2020. This raises the question whether public trust in science will remain on a high level as the pandemic endures. For an answer, it is helpful to place the results reported here within the context of results from the COSMO survey [4,18], which is a weekly online panel that asks about people's perceptions of COVID-19 in Germany which started in the first week of March 2020. These surveys provide a comprehensive overview of how participants experience and cope with the pandemic. With regard to trust in science, the most pertinent scale asks about people's trust in several institutions' capacity to deal "good and right" with the crisis (p. 46 ff.). The reported mean of trust in science in mid-April 2020, at the end of May 2020 and in mid-November 2020 was above the midpoint of the scale (around 5 on a scale ranging from 1 = "very low trust" to 7 = "very high trust"). Science (together with the medical system) belonged to those institutions trusted most. Political authorities (federal as well as state governments) were trusted less. Although still above the midpoint of the scale, the level of trust in the latter actors is gradually decreasing.

The Robert Koch Institute (RKI) is an interesting case here, because it is the federal reference agency for the provision of science-based information about the pandemic. In a sense, it is a kind of interface between science and politics. While trust in this institution is significantly higher than trust in the federal and state departments for health (*Gesundheitsministerium*), the decline of trust in this institution is similar to the decline of trust in political authorities.

Our conclusion: When it comes to public's general trust in science the *Science Barometer* surveys reported here indicate an increasing decoupling of trust in science and trust in politics. While *trust in politics* is decreasing, *trust in science* and the *endorsement of a science-based politics* remain on a high level. The predictive power of the endorsement of a science-based politics for trust in science is strong. If–in the long run–more and more German people feel that politicians do not fulfill this expectation, the decreasing trust in politics may result in a pull-down effect of trust in politics on trust in science. Trust in science might not only be a condition, it might also become a result of public perceptions of policies that are seemingly or actually based on scientific knowledge.

With regard to whether the stark increase of trust in science will persist, we would argue that science could remain an anchor in troubled times for the majority of German people, However, this will only last for as long as those navigating the ship make reasonable use of this tool.

## Limitations, strengths, and further directions

This paper is based on a secondary analysis of data available from existing surveys; meaning it is not a report of results from surveys designed exactly to answer its specific research questions. Therefore, we like to address several limitations.

First, we had to rely mainly on single-item measurements, such as the item for trust in science and research. However, we were able to corroborate our results with additional items that measured related aspects. A detailed report on results for single items has the advantage that results can be judged based on the exact wording of the survey questions. When summarizing results from different countries in the context of COVID-19, Askvall, Bucchi, Fähnrich, Trench and Weißkopf, p. 3, [58]) concluded: "When it comes to findings specifically on trust, the precise wording of the questions has a significant bearing".

Second, we only had a small set of variables included in the PreCOVID survey and across all COVID-19 related surveys. Only the within-pandemic surveys shared a larger common set of variables that allowed for further comparative conclusions.

Third, due to the fact that most German citizens had at least some trust in science, the variance of this variable was restricted (also shown by its skewed distribution). This may also be an explanation for why only a limited proportion of the belief variables actually related to the trust judgments. However, we still maintain, that the proportions of agreement with these further belief variables are generically interesting when considered in the bigger picture.

Fourth, there are some variables that one would expect to be related to trust in science (because of theoretical reasons or because of moderate zero-order correlations; S2 Table) but which are not significant predictors of trust in science in (some of) the regression analyses. One reason for this may be that the single effects of these variables on trust (even if they are significant) cancel each other out when being included in the same model due to their correlation (and hence due to minor multicollinearity). So, if we test the effects of two of these items on trust in science in the same regression model, we control for the influence of one item on the other and vice versa. Consequently, none of the items may reach significance even if they would as single predictors. This phenomenon is especially likely if one uses multiple predictors that are closely related (as it is in our case). This could lead to an "overfit" of the model [59]. To address this in the present study, our interpretations mostly relied on those regressions in which items were included for the first time. For example, the demographic variables were included in all analyses, but we have focused our interpretation of them on the first regression analysis including only a few other variables (Table 2).

Most of the other COVID-related surveys mentioned above were based on online panels. It is a strength of our paper that the surveys under scope were based on telephone interviews, which are less prone to self-selection effects and oversampling of well-educated respondents.

The availability of data from before and within the pandemic is clearly another strength of our analyses. Nevertheless, conclusions regarding the variability of beliefs have to be considered with due caution, since this variability could only be described on an aggregated level. There is a need for further research using repeated within-measurement. Such designs would allow for a more detailed analysis of the impact of person variables, such as education (our study), epistemic beliefs [47], political orientation (our study, [60]), ideologies [61], conspiracy beliefs [51], science-related populist attitudes [43] and moral concerns [8] on the variability of trust judgments.

It would also be worthwhile to scrutinize the interplay between peoples' views about science, about science-based politics and politics in greater detail. We suggest two possible approaches for this, which could also be combined:

a. A focus on trust in science and evaluations of the performance of different stakeholders in fighting the pandemic. This would require a much more detailed coverage of peoples' perceptions and assessments of the pandemic as well as of the measures taken for its containment, such as in the COSMO study mentioned above. The measures against the pandemic affect people differently, depending on context factors as their income and housing

situation. They might also be perceived differently with regard to issues of fairness and inequality. Simultaneously exploring these aspects along with trust in science at a high level of detail, as presented herein, would allow for modeling the relationship between trust in science and trust in politics. In combination with designs for measuring within-subject variability, (as, for example [10]), it would then be possible to test the pull-down hypotheses about backlash effects of science-based politics on trust in science.

b. A focus on the interplay between science and politics. For science-based politics scientific knowledge must somehow be transferred into the realm of politics. This includes the challenges of expert-layperson communication, because most politicians are laypersons, (just as most other citizens) with regard to the respective sciences. This also includes the challenges of transforming evidence about *what is* into recommendations *what to do*. Political measures must consider goals and norms as well as practical constraints. It is rather rare that scientific evidence immediately tells what to do on a political level. Scientists might become involved in the transforming of evidence into policies [62]. There are different ways for giving advice and these come along with different roles for scientists [63,64]. The Science Barometer surveys have seized the complexity of these different ways to some extent by asking separately about the endorsement of *science-based politics* and of *scientists active engagement in politics*. Our results regarding these two items support the conclusion (see above) that the public discerns these different kinds of relating science to politics. Digging deeper into the interplay between (trust in) science and (trust in) politics would require further items portraying the just sketched complexities of this interplay.

## Practical implications for science communication in the context of COVID-19 and beyond

If COVID-19 is a stress test for trust in science, as stated by an editorial in *The Lancet* [65] in autumn 2020, science in Germany, in general, has so far passed this test. The results presented herein are encouraging for scientists, as they can build on a strong credit of trust. Nevertheless, they also mark the possibility that overall trust in science may decline, opening gateways for the propagation of weird beliefs about science. Therefore, it is important to engage citizens in discussions about the process and contexts of knowledge production, including its uncertainties. The feeling of not understanding the science of COVID-19 negatively predicts trust in science. Therefore, it is crucial to explain whatever can be understood without an expert background, while simultaneously ensuring the public to avoid the *illusion of understanding* [49,66]. Emphasizing the complexities of COVID-19 as a research topic could help to attribute the *feelings of not understanding* the science of COVID-19 to the division of cognitive labor between citizens and scientists, rather than to a conspiracy that *scientists are not telling us all that they know*, for example. It might be helpful to discuss the reasons for controversies among scientists [67,68] as such controversies have the potential to be used as gateways for populist science denialism [69]. Since two-thirds of German people agree with the statement that *controversies among scientists are helpful*, this widely shared belief represents a good base for further explaining the role of controversies and other aspects in the process of knowledge production.

In the same vain, our results on reasons for trust and distrust suggest some topics for science communication that could support citizens' informed trust judgments [70]: People relied mainly on the *expertise*, less on the *integrity* and the least on the *benevolence* of scientists when it came to reasons for trusting them. When it came to reasons for distrust, an inverse pattern was found with the violation of expectations regarding the *benevolence* of scientists being the

most important reason for distrust. The COVID-19 pandemic requires the *expertise* of many disciplines (such as virology, epidemiology, psychology, economy, and sociology), and many issues of interest for citizens transgress the boundaries of one discipline.

Accordingly, it is important for supporting informed trust in science and in scientists to explain clearly a scientist's specific kind of *expertise* and its limitations regarding the pandemic. Further, it is important with regard to *integrity* as a further trust dimension to be explicit about the possibilities and the limitations of scientists for the design and implementation of science-based policy to cope with the pandemic. Scientists as well as scientific organization should not only talk about their findings regarding the pandemic, they should also provide information about their roles (in relation to the roles of other stakeholders) when it comes to developing and implementing policies for fighting the pandemic. When the general publics' trust in politics is decreasing it becomes even more important for scientists to emphasize their independence from governments and politicians [10].

When it comes to countering attacks that aim to sow distrust, it might be more important to emphasize the intention, interests, and values [62] of scientists. Supporting citizens' informed judgements about the benevolence of scientists but also about the (lack of) benevolence of those who aim to erode trust in science could help to maintain trust in science as an anchor when coping with the challenges of a pandemic [71].

## Supporting information

**S1 Fig. trustsci_freq.**
(TIFF)

**S2 Fig. poldec_freq.**
(TIFF)

**S3 Fig. scpo_interfpolit_freq.**
(TIFF)

**S4 Fig. pvar_trustpolitician_freq.**
(TIFF)

**S5 Fig. pvar_trustjournalist_freq.**
(TIFF)

**S6 Fig. pvar_trustpeers_freq.**
(TIFF)

**S7 Fig. meti_trustexp_freq.**
(TIFF)

**S8 Fig. meti_trustint_freq.**
(TIFF)

**S9 Fig. meti_trustben_freq.**
(TIFF)

**S10 Fig. meti_distrexp_freq.**
(TIFF)

**S11 Fig. meti_distrint_freq.**
(TIFF)

**S12 Fig. meti_distrben_freq.**
(TIFF)

**S13 Fig. pvar_wllngvax_freq.**
(TIFF)

**S14 Fig. perc_controversy_freq.**
(TIFF)

**S15 Fig. perc_intllctlhum_freq.**
(TIFF)

**S16 Fig. perc_complicated_freq.**
(TIFF)

**S17 Fig. bpos_stopspread_freq.**
(TIFF)

**S18 Fig. bpos_vaccinedev_freq.**
(TIFF)

**S19 Fig. bpos_notunderst_freq.**
(TIFF)

**S20 Fig. scep_concealmnt_freq.**
(TIFF)

**S21 Fig. scep_infosource_freq.**
(TIFF)

**S22 Fig. scep_exaggerate_freq.**
(TIFF)

**S23 Fig. scep_noevidence_freq.**
(TIFF)

**S24 Fig. scpo_commonsense_freq.**
(TIFF)

**S25 Fig. pvar_risk_likcontract_freq.**
(TIFF)

**S26 Fig. pvar_risk_likseverity_freq.**
(TIFF)

**S27 Fig. pvar_approvemeasures_freq.**
(TIFF)

**S1 Table. Descriptive statistics for *trust in science and research* in sample subgroups.**
(DOCX)

**S2 Table. Comparisons between surveys (U-tests).**
(DOCX)

**S3 Table. Zero-order correlation of predictors regarding RQ 2 with trust in science and research.**
(DOCX)

**S4 Table. Predicting changes in *trust in science and research* before and during the Covid-19 pandemic.**
(DOCX)

**S5 Table. Predicting changes in *trust in science and research* before and during the Covid-19 pandemic focusing on beliefs in science measured in 04/2020 and 11/2020.**
(DOCX)

**S6 Table. Predicting trust in science with additionally focusing beliefs in science (04/2020), respectively sceptical attitudes and trust/distrust reasons (11/2020).**
(DOCX)

## Acknowledgments

Thanks to Felia Kochanek for supporting the reporting about the COVID-19 situation in Germany. Thanks to Jana Michels for supporting the design of S2 Table.

## Author Contributions

**Conceptualization:** Rainer Bromme, Niels G. Mede, Eva Thomm, Bastian Kremer, Ricarda Ziegler.

**Data curation:** Rainer Bromme, Niels G. Mede, Eva Thomm, Bastian Kremer, Ricarda Ziegler.

**Formal analysis:** Niels G. Mede, Eva Thomm, Bastian Kremer.

**Investigation:** Bastian Kremer, Ricarda Ziegler.

**Methodology:** Rainer Bromme, Niels G. Mede, Eva Thomm, Ricarda Ziegler.

**Project administration:** Rainer Bromme.

**Resources:** Bastian Kremer, Ricarda Ziegler.

**Software:** Niels G. Mede.

**Writing – original draft:** Rainer Bromme.

**Writing – review & editing:** Rainer Bromme, Niels G. Mede, Eva Thomm, Bastian Kremer, Ricarda Ziegler.

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
