## [Decision Letter · Decision Letter 0]

9 Sep 2021

PONE-D-21-21482An anchor in troubled times: Trust in science before and within the COVID-19 pandemicPLOS ONE

Dear Dr. Bromme,

Thank you for submitting your manuscript to PLOS ONE. After careful consideration, we feel that it has merit but does not fully meet PLOS ONE’s publication criteria as it currently stands. Therefore, we invite you to submit a revised version of the manuscript that addresses the points raised during the review process.

We look forward to receiving your revised manuscript.

Kind regards,

Prof. Anat Gesser-Edelsburg, Ph.D.

Academic Editor

PLOS ONE

Journal Requirements:

3. We noted in your submission details that a portion of your manuscript may have been presented or published elsewhere. [This paper compares data from 4 surveys, data which are available upon request from WID.

From our group of authors NM and RZ (see below) were involved in an analysis which made use of one of the 4 data sets, but which had a different focus. 

Nothing (exept of the survey data) of the Mede et al 2020 paper was used in our paper.] Please clarify whether this conference proceeding or publication was peer-reviewed and formally published. If this work was previously peer-reviewed and published, in the cover letter please provide the reason that this work does not constitute dual publication and should be included in the current manuscript.

Reviewers' comments:

Reviewer's Responses to Questions

**Comments to the Author**

1. Is the manuscript technically sound, and do the data support the conclusions?

Reviewer #1: Yes

Reviewer #2: Yes

2. Has the statistical analysis been performed appropriately and rigorously? 

Reviewer #1: I Don't Know

Reviewer #2: Yes

3. Have the authors made all data underlying the findings in their manuscript fully available?

Reviewer #1: Yes

Reviewer #2: Yes

4. Is the manuscript presented in an intelligible fashion and written in standard English?

Reviewer #1: Yes

Reviewer #2: Yes

5. Review Comments to the Author

Reviewer #1: General Comments:

The results presented in the paper “An anchor in troubled times: Trust in science before and within the COVID-19 pandemic” are based on a secondary analysis of data obtained from existing surveys and evaluates the perceptions of people in Germany during a one-year period of the pandemic Sept 19-Nov 20. This data from four cross-sectional survey waves presents interesting findings on how the public’s perception of science changed as the pandemic progressed. The authors presented a straightforward analysis demonstrating the data supported the conclusions that in general there was trust in science, an endorsement of science-based politics, and that trust is related to expectations. The limitations were presented clearly and what appears to be a robust statistical analysis supports the findings. Evaluating what perceived factors account for the variability was also competed and the regressions models that looked at predictors brought forth interesting conclusions. The authors did an excellent job in their organization of the paper and the presentation of the findings in both the results and the discussion sections.

Specific Comments:

On page 30, could the statement in lines 626-628 “The magnitude of trust in science and in scientists as well as respondents’ high expectation for science to solve problems need to be considered in a broader context that encompasses the role of politics as another, even more crucial factor for effectively coping with the pandemic” be expanded. I find this to be one of the key conclusions and would like the authors to share more on their thoughts (here or later in the discussion). There is some literature out there already and add some context would be beneficial. Similarly, I would like to see some discussion added on the role of wealth and equity inequality, a topic not included here, be mentioned within the discussion as this again is another area worth mentioning, for future studies in Germany but more so as very relevant for other countries where such factors are more critical.

Finally, I would like to thank the authors for this publication. By publishing the data for Germany, one would hope that yes, this promotes other countries to follow since clearly the issues around trust and distrust in science and the need for more open discussion as demonstrated in this manuscript are real. Understanding the complex interactions between scientific trust, politics and policy are key to the development of global policies for the future and management of future pandemics.

Reviewer #2: I appreciated having the opportunity to review this manuscript, this is a timely and informative study. I applaud the authors for their work, which really surprised me for their intensive analysis.

Comment 1: In the RESULTS part 1, all the analysis results were reported in the forms of text, maybe several Tables or Figures to illustrate those results would be much more comfortable for the readers. Hence, my suggestion is just make the results into tables/figures if possible.

Comment 2: I noticed that across the manuscript, especially in Table 2, Table 3, the DV is conceptualized using abbreviations, while many IVs are in the format of wording in questionnaires. For instance, “Political decisions should be based on scientific evidence”, which could be named as endorsement of science-based politics.

Comment 3: I was surprised at Table 3, among 15 predictors, only about three or five are significantly associated with Trust in Science. Moreover, some typical and critical factors like education level (a proxy of scientific knowledge) are not significantly associated with the Trust in science on the 04/2020 wave. Besides, some of scientific/scientists belief related predictors, including “Science and research on Corona are so complicated that I do not understand much of it”, “Most scientists currently speaking up differentiate clearly between what they know for sure and what are open questions on Corona”, “It is not up to scientists to get involved in politics.”, “We should rely more on common sense when dealing with corona and we do not need any scientific studies for this”, are not significantly associated with Trust in science, seems weird and inconsistent with the previous studies. It would be much appreciated if you discuss that in the discussion PART.

6. PLOS authors have the option to publish the peer review history of their article (what does this mean?). If published, this will include your full peer review and any attached files.

Reviewer #1: No

Reviewer #2: No

---

## [Author Response · Author response to Decision Letter 0]

13 Dec 2021

We would like to thank you and the reviewers for supportive and encouraging feedback. It helped us to revise and improve our manuscript, which we have re-submitted just now. In the following, we have specified they ways in which we revised our contribution in order to address your and the reviewers’ concerns. 

Done

Yes, we see now that the former notification was confusing. The following 'Funding Information' (now inserted in the Funding information section) should be more precise. 

Funding Information:

Wissenschaft im Dialog (WiD, "Science in Dialogue") is a German non-profit organization for science communication. Organizing and reporting the annual survey "Wissenschaftsbarometer" (Science Barometer) survey is one of its regular projects. The surveys are funded by the Robert Bosch Stiftung (www.bosch-stiftung.de, grant number: 00906101-0030) and the Fraunhofer-Gesellschaft (www.fraunhofer.de, no grant number available). Neither the funders of the Wissenschaftsbarometer nor the general funders of Wissenschaft im Dialog were involved in the present re-analysis of the survey data.

Co-authors Bastian Kremer and Ricarda Ziegler contributed to this paper as employees of Wissenschaft im Dialog and, therefore, their worktime for the analyses presented in this paper is subject to the funding provided by Robert Bosch Stiftung and Fraunhofer-Gesellschaft. Apart from that, none of the authors received specific funding for their work on this paper. Rainer Bromme is a member of the international scientific advisory board that advises on the design of the annual survey (Wissenschaftsbarometer). The advisory board was not involved in the present re-analysis of the survey data.

3. We noted in your submission details that a portion of your manuscript may have been presented or published elsewhere. [This paper compares data from 4 surveys, data which are available upon request from WID. From our group of authors NM and RZ (see below) were involved in an analysis which made use of one of the 4 data sets, but which had a different focus. Nothing (exept of the survey data) of the Mede et al 2020 paper was used in our paper.] Please clarify whether this conference proceeding or publication was peer-reviewed and formally published. If this work was previously peer-reviewed and published, in the cover letter please provide the reason that this work does not constitute dual publication and should be included in the current manuscript.

Due to a misunderstanding within the group of authors, the above cited notification was simply false. Actually, contributors NM and RZ have co-published a peer-reviewed journal article that relies on the 2018 wave of the Science Barometer Germany and focused on the German public’s awareness and perceptions of the “replication crisis”:

Mede, N. G., Schäfer, M. S., Ziegler, R., & Weißkopf, M. (2020). The “replication crisis” in the public eye: Germans’ awareness and perceptions of the (ir)reproducibility of scientific research. Public Understanding of Science. doi:10.1177/0963662520954370

The present study only uses data from 2019 and 2020 and has a very different focus, hence it does not constitute dual publication. We deleted the entry within the submission form, entering now: NO

We have deleted the reference: 

Kaltenborn KF. Good science in Zeiten der Coronavirus-Pandemie. Deutschland in der zweiten Pandemie-Welle. wwwb-i-t-onlinede [Internet]. 2021; 24(1).

because it was no longer available (for example, via Research Gate), but we did not find a retraction note.

In response to reviewer requests and for completion of our literature review, we have added the following references:

Algan, Y., Cohen, D., Davoine, E., Foucault, M., & Stantcheva, S. (2021). Trust in scientists in times of pandemic: Panel evidence from 12 countries. Proceedings of the National Academy of Sciences, 118(40), e2108576118. doi:10.1073/pnas.2108576118

Bromme, R. (2020). Informiertes Vertrauen: Eine psychologische Perspektive auf Vertrauen in Wissenschaft [Informed trust in science: A psychological perspective on trust in science]. In M. Jungert, A. Frewer, & E. Mayr (Eds.), Wissenschaftsreflexion. Interdisziplinäre Perspektiven zwischen Philosophie und Praxis [Reflections on Science. Interdisciplinary perspectives between philosophy and experience] (pp. 105-134). Paderborn: Mentis.

Eberl, J.-M., Huber, R. A., & Greussing, E. (2021). From populism to the “plandemic”: why populists believe in COVID-19 conspiracies. Journal of Elections, Public Opinion and Parties, 31(sup1), 272-284. doi:10.1080/17457289.2021.1924730

Evans, J. H., & Hargittai, E. (2020). Who Doesn’t Trust Fauci? The Public’s Belief in the Expertise and Shared Values of Scientists in the COVID-19 Pandemic. Socius, 6. doi:10.1177/2378023120947337

Lever, J., Krzywinski, M., & Altman, N. (2016). Model selection and overfitting. Nature Methods, 13(9), 703-704. doi:10.1038/nmeth.3968

Mede, N. G., & Schäfer, M. S. (2021). Science-related populism declining during the COVID-19 pandemic: A panel survey of the Swiss population before and after the Coronavirus outbreak. Public Understanding of Science. doi:10.1177/09636625211056871

Pielke, R. J. (2007). The Honest Broker: Making Sense of Science in Policy and Politics. Cambridge: Cambridge University Press.

Rudert, S., Gleibs, I. H., Gollwitzer, M., Hajek, K., S, Harth, N., A. , Häusser, J., . . . Schneider, D. (2021). Us and the virus: understanding the COVID-19 pandemic through a social psychological lens. European Psychologist. Retrieved from http://eprints.lse.ac.uk/111012/

Sokolovska, N., Fecher, B., & Wagner, G. (2019). Communication on the Science-Policy Interface: An Overview of Conceptual Models. Publications, 7(4), 1-15. doi:10.3390/publications7040064

Travis, J., Harris, S., Fadel, T., & Webb, G. (2021). Identifying the determinants of COVID-19 preventative behaviors and vaccine intentions among South Carolina residents. PLOS ONE, 16(8), e0256178. doi:10.1371/journal.pone.0256178

Furthermore, we updated the reference list and made sure that (whenever available) each entry includes either a "doi…." or a "available at http…..".

3. Have the authors made all data underlying the findings in their manuscript fully available?

In the previous version, raw data and syntax were available via two different repositories. We have now put everything together in one repository, thereby making all files accessible without restriction. We included the link to the OSF project within the manuscript in order to adhere PLOS ONE’s requirements: 

The analyses could be reproduced with the R syntax, which we share publicly with the complete survey data and original questionnaires at https://osf.io/czn4g/?view_only=ae4fe4eac8564077b3bb6fe68844f745

4. Is the manuscript presented in an intelligible fashion and written in standard English?

We have carefully revised the full paper, resulting in many changes of the wording, sometimes of the syntax and of the flow of arguments (See TRACKED file). Finally, we have sent it to a professional English language editing service (SCRIBENDI), resulting in further changes. We asked the SCRIBENDI editor to take care that standard American English was consistently used.

Reviewer #1: General Comments: 

The results presented in the paper “An anchor in troubled times: Trust in science before and within the COVID-19 pandemic” are based on a secondary analysis of data obtained from existing surveys and evaluates the perceptions of people in Germany during a one-year period of the pandemic Sept 19-Nov 20. This data from four cross-sectional survey waves presents interesting findings on how the public’s perception of science changed as the pandemic progressed. The authors presented a straightforward analysis demonstrating the data supported the conclusions that in general there was trust in science, an endorsement of science-based politics, and that trust is related to expectations. The limitations were presented clearly and what appears to be a robust statistical analysis supports the findings. Evaluating what perceived factors account for the variability was also competed and the regressions models that looked at predictors brought forth interesting conclusions. The authors did an excellent job in their organization of the paper and the presentation of the findings in both the results and the discussion sections.

Many thanks for these encouraging comments.

Specific Comments: 

5. On page 30, could the statement in lines 626-628 “The magnitude of trust in science and in scientists as well as respondents’ high expectation for science to solve problems need to be considered in a broader context that encompasses the role of politics as another, even more crucial factor for effectively coping with the pandemic” be expanded. I find this to be one of the key conclusions and would like the authors to share more on their thoughts (here or later in the discussion). There is some literature out there already and add some context would be beneficial. 

Similarly, I would like to see some discussion added on the role of wealth and equity inequality, a topic not included here, be mentioned within the discussion as this again is another area worth mentioning, for future studies in Germany but more so as very relevant for other countries where such factors are more critical.

Thank you for this comment! We followed this suggestion in four ways: 

First, we reconsidered our findings, especially the fact that trust in science has declined only a little, while trust in politics decreased more severely. When reporting results about trust in science compared to trust in politics, we (hopefully) made the decoupling of both kinds of trust between April and November 2020 more clear. 

Furthermore, we elaborated on the problems this development might bear, introducing the notion of a pull-down effect of trust in politics on trust in science. In the last two paragraphs of the discussion section The sustainability of the present trust in science we have added:

Lines 783 794

Our conclusion: When it comes to public's general trust in science the Science Barometer surveys reported here indicate an increasing decoupling of trust in science and trust in politics. While trust in politics is decreasing, trust in science and the endorsement of a science-based politics remain on a high level. The predictive power of the endorsement of a science-based politics for trust in science is strong. If – in the long run – more and more German people feel that politicians do not fulfill this expectation, the decreasing trust in politics may result in a pull-down effect of trust in politics on trust in science. Trust in science might not only be a condition, it might also become a result of public perceptions of policies that are seemingly or actually based on scientific knowledge. 

With regard to whether the stark increase of trust in science will persist, we would argue that science could remain an anchor in troubled times for the majority of German people, However, this will only last for as long as those navigating the ship make reasonable use of this tool. 

We also added more references (see above), which either had a comparative focus (for example, Algan et al.) or support our elaboration of the science-politics relationship. 

Secondly, we elaborated the science policy relationship in the Limitations/Future directions section because we think that the specificities of this relationship are a core feature of 'context'.

Lines 843-867

It would also be worthwhile to scrutinize the interplay between peoples' views about science, about science-based politics and politics in greater detail. We suggest two possible approaches for this, which could also be combined:

a) A focus on trust in science and evaluations of the performance of different stakeholders in fighting the pandemic. This would require a much more detailed coverage of peoples' perceptions and assessments of the pandemic as well as of the measures taken for its containment, such as in the COSMO study mentioned above. The measures against the pandemic affect people differently, depending on context factors as their income and housing situation. They might also be perceived differently with regard to issues of fairness and inequality. Simultaneously exploring these aspects along with trust in science at a high level of detail, as presented herein, would allow for modeling the relationship between trust in science and trust in politics. In combination with designs for measuring within-subject variability, (as, for example (Algan, Cohen, Davoine, Foucault, & Stantcheva, 2021)), it would then be possible to test the pull-down hypotheses about backlash effects of science-based politics on trust in science.

b) A focus on the interplay between science and politics. For science-based politics scientific knowledge must somehow be transferred into the realm of politics. This includes the challenges of expert-layperson communication, because most politicians are laypersons, (just as most other citizens) with regard to the respective sciences. This also includes the challenges of transforming evidence about what is into recommendations what to do. Political measures must consider goals and norms as well as practical constraints. It is rather rare that scientific evidence immediately tells what to do on a political level. Scientists might become involved in the transforming of evidence into policies (Evans & Hargittai, 2020). There are different ways for giving advice and these come along with different roles for scientists (Pielke, 2007; Sokolovska, Fecher, & Wagner, 2019). The Science Barometer surveys have seized the complexity of these different ways to some extent by asking separately about the endorsement of science-based politics and of scientists active engagement in politics. Our results regarding these two items support the conclusion (see above) that the public discerns these different kinds of relating science to politics. Digging deeper into the interplay between (trust in) science and (trust in) politics would require further items portraying the just sketched complexities of this interplay.

Thirdly, (and in the same vain as the preceding section) we added a further paragraph to the final section on Practical implications for science communication:

Lines 896 -910

Accordingly, it is important for supporting informed trust in science and in scientists to explain clearly a scientist’s specific kind of expertise and its limitations regarding the pandemic. Further, it is important with regard to integrity as a further trust dimension to be explicit about the possibilities and the limitations of scientists for the design and implementation of science-based policy to cope with the pandemic. Scientists as well as scientific organization should not only talk about their findings regarding the pandemic, they should also provide information about their roles (in relation to the roles of other stakeholders) when it comes to developing and implementing policies for fighting the pandemic. When the general publics' trust in politics is decreasing it becomes even more important for scientists to emphasize their independence from governments and politicians(Algan et al., 2021).

When it comes to countering attacks that aim to sow distrust, it might be more important to emphasize the intention, interests, and values (Evans & Hargittai, 2020) of scientists. Supporting citizens' informed judgements about the benevolence of scientists but also about the (lack of) benevolence of those who aim to erode trust in science could help to maintain trust in science as an anchor when coping with the challenges of a pandemic (Lewandowsky & van der Linden, 2021). 

With regard to wealth: Within the suggestions for further research, we inserted: 

Lines 846-848

The measures against the pandemic affect people differently, depending on context factors as their income and housing situation. They might also be perceived differently with regard to issues of fairness and inequality.

We are aware that this is not a deep coverage of the wealth /inequality issues, but we do not see how to go more into this direction within the context of our paper. 

6. Finally, I would like to thank the authors for this publication. By publishing the data for Germany, one would hope that yes, this promotes other countries to follow since clearly the issues around trust and distrust in science and the need for more open discussion as demonstrated in this manuscript are real. Understanding the complex interactions between scientific trust, politics and policy are key to the development of global policies for the future and management of future pandemics.

Thank you for this comment. This comment was inspiring for writing the above cited paragraphs with suggestions for further research. 

In the same vain of your comment is the final sentence of the first page (Introduction, line 67/68):

We report on the situation in Germany as an interesting case (and hope that this will provide motivation for a series of comparable reports in other countries).

Reviewer #2:I appreciated having the opportunity to review this manuscript, this is a timely and informative study. I applaud the authors for their work, which really surprised me for their intensive analysis.

Thank you for this comment

8. Comment 1: In the RESULTS part 1, all the analysis results were reported in the forms of text, maybe several Tables or Figures to illustrate those results would be much more comfortable for the readers. Hence, my suggestion is just make the results into tables/figures if possible.

We did as suggested: All U-test statistics as well as the means were put into Supplement Table 2, which resulted in a substantial shortening of the results section regarding Results Part 1. (Lines -282-43). With regard to the relative response frequencies, (contrasting the top-two levels of agreement with the bottom-two levels of agreement) we decided to keep these percentages into the text body. Transferring these figures into a further table would have required paraphrasing the percentages in order to report the main results within the text.

Inspired by your comment we also re-organized the vertical sequence of all belief variables in all tables. They are now in accordance with the flow of reporting in the results section of RQ1 (with an exception in table S 6, in order to make immediately visible that the last block of items (reasons for trust) was surveyed only in 11/20). Furthermore, we added more references to tables than we had in the previous version. 

9. Comment 2: I noticed that across the manuscript, especially in Table 2, Table 3, the DV is conceptualized using abbreviations, while many IVs are in the format of wording in questionnaires. For instance, “Political decisions should be based on scientific evidence”, which could be named as endorsement of science-based politics.

We have standardized the exact wording describing the variables in all tables. Within the text, when a belief variable is introduced, we used the wording of the questionnaire and later, when possible, a naming like the suggested one, for example endorsement of science-based politics or expertise, integrity and benevolence for naming the items on reasons for trusting/distrusting scientists. However, for the majority of belief items a shortened version of the original wording is less laborious to remember than a new wording, therefore in most cases we chose this option. Within the tables, we shortened the original wording whenever necessary for avoiding more than two lines within a cell.

10. Comment 3: I was surprised at Table 3, among 15 predictors, only about three or five are significantly associated with Trust in Science. Moreover, some typical and critical factors like education level (a proxy of scientific knowledge) are not significantly associated with the Trust in science on the 04/2020 wave. Besides, some of scientific/scientists belief related predictors, including “Science and research on Corona are so complicated that I do not understand much of it”, “Most scientists currently speaking up differentiate clearly between what they know for sure and what are open questions on Corona”, “It is not up to scientists to get involved in politics.”, “We should rely more on common sense when dealing with corona and we do not need any scientific studies for this”, are not significantly associated with Trust in science, seems weird and inconsistent with the previous studies. It would be much appreciated if you discuss that in the discussion PART.

You raise a point that needed further discussion – so thank you for bringing it up! 

In the section Limitations and Further Directions, this is how we discussed it now: 

Lines 816-829

Fourth, there are some variables that one would expect to be related to trust in science (because of theoretical reasons or because of moderate zero-order correlations; Table S 2) but which are not significant predictors of trust in science in (some of) the regression analyses. One reason for this may be that the single effects of these variables on trust (even if they are significant) cancel each other out when being included in the same model due to their correlation (and hence due to minor multicollinearity). So, if we test the effects of two of these items on trust in science in the same regression model, we control for the influence of one item on the other and vice versa. Consequently, none of the items may reach significance even if they would as single predictors. This phenomenon is especially likely if one uses multiple predictors that are closely related (as it is in our case). This could lead to an “overfit” of the model (Lever, Krzywinski, & Altman, 2016). To address this in the present study, our interpretations mostly relied on those regressions in which items were included for the first time. For example, the demographic variables were included in all analyses, but we have focused our interpretation of them on the first regression analysis including only a few other variables (Table 2).

We also examined your remark empirically: We calculated zero-order correlations of trust in science and each individual predictor (see Table S 3) and referred to this table in the beginning of the results section about RQ 2. These correlations do not control for potential effects of other predictors – and then your assumption is true.

With regard to education as a predictor, it is also important to note that it is predictive for trust in science in all surveys (see Table 2), but that its impact has been stronger after one year pandemic than it was before. Inspired by your comment we emphasized this by rewriting the first paragraph of the Discussion RQ 2 section: 

Lines 686-703

One striking finding was the stark increase of explained variance in trust in science from before to during the pandemic. This was evident even within the first set of our analyses which only included a few critical demographic variables and an endorsement of science-based politics item. This suggests that respondents' considerations about trust in science were more immersed in their personal reality, assuming that the demographic variables (such as education) actually mirrored this personal reality. The COVID-19 pandemic affected everybody's lives, rendering science, (presumably: health-related disciplines), more salient overall. When asked about trust in science, respondents probably interpreted the survey question more personally than respondents in previous surveys, and personal factors may have mattered more for their trust judgments. Our conclusion is illustrated by the variable education. While higher education was associated with trust in science in all surveys (see Table 2), this effect appeared strongest in 11/20. Above this, education also predicted the increase of trust between the PreCOVID survey in 9/19 and the survey 14 months later (11/20). 

This finding aligns with the above-discussed finding regarding respondents’ decrease in their feeling of understanding, which negatively predicted trust in science. At the beginning of the pandemic, most people felt that they had learned something about the virus and its effects. However, as the pandemic persisted, it became obvious that many issues around COVID-19 were rather complex, but having a better education could help respondents to cope with this complexity. 

Thanks again for your and for the reviewers' comments which we took as encouragement to revise the manuscript thoroughly.

---

## [Decision Letter · Decision Letter 1]

6 Jan 2022

An anchor in troubled times: Trust in science before and within the COVID-19 pandemic

PONE-D-21-21482R1

Dear Dr. Bromme,

We’re pleased to inform you that your manuscript has been judged scientifically suitable for publication and will be formally accepted for publication once it meets all outstanding technical requirements.

Kind regards,

Prof. Anat Gesser-Edelsburg, Ph.D.

Academic Editor

PLOS ONE

Additional Editor Comments (optional):

Reviewers' comments:

Reviewer's Responses to Questions

**Comments to the Author**

1. If the authors have adequately addressed your comments raised in a previous round of review and you feel that this manuscript is now acceptable for publication, you may indicate that here to bypass the “Comments to the Author” section, enter your conflict of interest statement in the “Confidential to Editor” section, and submit your "Accept" recommendation.

Reviewer #1: All comments have been addressed

2. Is the manuscript technically sound, and do the data support the conclusions?

Reviewer #1: (No Response)

3. Has the statistical analysis been performed appropriately and rigorously? 

Reviewer #1: (No Response)

4. Have the authors made all data underlying the findings in their manuscript fully available?

Reviewer #1: (No Response)

5. Is the manuscript presented in an intelligible fashion and written in standard English?

Reviewer #1: (No Response)

6. Review Comments to the Author

Reviewer #1: (No Response)

7. PLOS authors have the option to publish the peer review history of their article (what does this mean?). If published, this will include your full peer review and any attached files.

Reviewer #1: No

---

## [Editor Report · Acceptance letter]

14 Jan 2022

PONE-D-21-21482R1 

An anchor in troubled times: Trust in science before and within the COVID-19 pandemic 

Dear Dr. Bromme:

I'm pleased to inform you that your manuscript has been deemed suitable for publication in PLOS ONE. Congratulations! Your manuscript is now with our production department. 

Kind regards, 

on behalf of

Prof. Anat Gesser-Edelsburg 

Academic Editor

PLOS ONE